# Experimental Study on Recentering Behavior of Precompressed Polyurethane Springs

**DOI:** 10.3390/ma15103514

**Published:** 2022-05-13

**Authors:** Young-Hun Ju, Iman Mansouri, Jong-Wan Hu

**Affiliations:** 1Department of Civil and Environmental Engineering, Incheon National University, Incheon 22012, Korea; wndudgns36@gmail.com; 2Incheon Disaster Prevention Research Center, Incheon National University, Incheon 22012, Korea; 3Department of Civil Engineering, Birjand University of Technology, Birjand 97175-569, Iran; mansouri@birjandut.ac.ir

**Keywords:** polyurethane spring, recentering force, smart materials, cyclic loading test, precompression

## Abstract

Traditional seismic design has a limitation in that its performance is reduced by significant permanent deformation after plastic behavior under an external load. The recentering characteristics of smart materials are considered to be a means to supplement the limitations of conventional seismic design. In general, the recentering characteristics of smart materials are determined by their physical properties, whereas polyurethane springs can regulate the recentering characteristics by controlling the precompression strain. Therefore, in this study, 160 polyurethane spring specimens were fabricated with compressive stiffness, specimen size, and precompression strain as design variables. The compression behavior and precompression behavior were studied by performing cyclic loading tests on a polyurethane spring. The maximum stress and maximum strain of the polyurethane spring showed a linear relationship. Precompression and recentering forces have an almost perfect linear relationship, and the optimal level of precompression at which residual strain does not occur was derived through regression analysis. Additionally, a prediction model for predicting recentering force based on the linear relationship between precompression and recentering force was presented.

## 1. Introduction

Existing seismic designs, such as isolators and dampers have a limitation in that recovery is infeasible or difficult in the event of damage or permanent deformation. This results in substantial repair/reinforcement costs [1,2,3,4]. For example, seismic isolation bearings and damper systems mainly utilize the principle of dissipating energy and the nonlinear behavior of steel [5,6]. Therefore, once the steel is subjected to nonlinear behavior, it loses its original properties, and its energy dissipation capacity is reduced significantly. As a result, it becomes nonreusable and thereby, needs to be replaced. This problem can be solved by providing additional restoring force and energy dissipation capacity using a member in which smart materials are applied to the part where plastic deformation is concentrated in the structure [7,8,9].

Smart materials can be defined as materials that can regain their original shape through physical and chemical stimuli, and this behavior is defined as smart behavior [10,11]. Representative smart materials include piezoelectric materials, shape memory alloys, and electro-rheostat materials [12,13,14,15,16]. In particular, research using superelastic shape memory alloys is being conducted actively in the seismic field.

Superelastic shape memory alloys are materials that are marginally different in strength from general steel. They have remarkable energy dissipation capabilities and can be restored to their original shape only by heat treatment or stress removal at room temperature [17,18]. However, superelastic shape memory alloys have disadvantages in that these are uneconomical to be commercialized owing to their high price, and the material specifications are limited because these are difficult to process. In addition, because the restoration performance is determined by the physical properties of the material, it can be applied only in a limited environment, and precise control is difficult.

Recently, polyurethane elastomers have attracted attention as materials that can replace superelastic shape memory alloys. Polyurethane elastomers are characterized in that they can be operated at temperatures below 0 °C and above 80 °C [19,20]. Polyurethane elastomers have remarkable anti-abrasion, corrosion resistance, and restoring forces. In addition, polyurethane elastomers possess excellent mechanical characteristics including high vertical strength and suitable fatigue life [21]. Therefore, it is being used in a wide variety of industries, such as automobiles, footwear, and electricity industries in addition to the construction industry [22,23,24]. A notable feature of polyurethane elastomers in the field of civil engineering is the recentering force that occurs under precompression conditions. Polyurethane elastomers display a certain level of recentering force by generating an internal force corresponding to the compressed strain when it is maintained in a compressed state by applying a compressive force initially. This behavior is similar to flag-shaped behavior owing to the shape memory effect of superelastic shape memory alloy [25].

In this study, the compression behavior and recentering force of polyurethane elastomer springs made under precompression conditions were analyzed through cyclic loading tests. The originality of this study is that it analyzed the relationship between the recentering force and precompression strain of a polyurethane spring and proposed a prediction model for the recentering force. The prediction model proposed in this study is expected to be able to predict the recentering force in an intuitive way according to the experimental results. Therefore, it has a significant difference from the existing complex hyperelastic behavior prediction equations. A total of 160 polyurethane spring specimens and a compression jig to generate precompression were manufactured. The overall precompression behavior characteristics of the polyurethane spring were verified through a cyclic loading test. Based on the experimental results, this study attempted to verify the applicability of the polyurethane spring as a compression member by analyzing the maximum stress, recentering force, prediction model of the recentering force, and energy dissipation capacity.

## 2. Design of Polyurethane Springs

Polyurethane springs are used mainly as compression members owing to their strong restoring force against compression loads. Therefore, the specimen was designed as a cylinder considering its applicability as a compression member (see Figure 1). Polyurethane has a hardness of 20–80 Shore D. That is, it has a fairly large hardness and can resist a relatively large load considering that general natural rubber has a hardness between 50 and 70 Shore A [26]. To prevent the specimen from being separated from the jig during the experiment, it was manufactured in the form of a cylinder with a hole having an inner diameter (Din, mm) of 20 mm in the center. In addition, the outer diameter (D, mm) ranged from 100 mm to 72 mm, and the length (L, mm) was designed to range from 100 mm to 60 mm.

A total of 160 experimental specimens were produced in this study. Each of these were named according to its length (L, mm), outer diameter (D, mm), precompression strain ($\varepsilon_{P}$, %), and maximum strain ($\varepsilon_{max}$, %) in that order (see Figure 2). For example, the specimen with a length, outer diameter, precompression strain, and maximum strain of 100 mm, 100 mm, 14%, and 30%, respectively, was designated as 100 L-100D-14P-30%. Therefore, model identification (ID) was assigned to individual cases using the procedure mentioned above, and cyclic loading tests were performed on the specimens according to the experimental condition from ID information.

The outer diameter and length of the polyurethane spring specimens were designed based on the compressive stiffness ($k_{c}$). The compressive stiffness can be calculated using Equation (1):(1)$$k_{c}=\frac{E_{a}A}{L}$$

Here, $k_{c}$ is the compressive stiffness, $E_{a}$ is the apparent elastic modulus of the polyurethane spring, A means the sectional area, and L means the length of the specimen. $E_{a}$ is calculated using Equation (2). It is composed of the shape modulus (S) and modulus of elasticity ($E_{o}$). In this study, a polyurethane spring with a modulus of elasticity of 68.95 MPa was used [27].
(2)$$E_{a}=E_{o}\left(1.2+2S^{2}\right)$$

The shape factor (S) can be calculated using Equation (3). It is expressed as the ratio of the area directly subjected to load to the area not receiving load [28]:(3)$$S=\frac{\mathrm{loaded}\;\mathrm{area}}{\mathrm{force}\;\mathrm{free}\;\mathrm{area}}$$

The specimen size of the polyurethane spring can be calculated based on the compressive-stiffness design in Equations (1)–(3) by varying the length and cross-sectional area while maintaining the compressive stiffness constant. In this study, for the convenience of specimen production, the size of the specimen with an identical compressive stiffness was adjusted by calculating the cross-sectional area corresponding to the length of each specimen by fixing the length to 100, 90, 80, 70, and 60 mm in Equation (1). Finally, precompression is a preprocessing step to generate the recentering force of the polyurethane spring. The precompression strain ($\varepsilon_{P}$) is defined as the ratio of the total specimen length to the precompression displacement:(4)$$\varepsilon_{P}=\frac{\mathrm{Displacement}\;\mathrm{of}\;\mathrm{precompression}}{\mathrm{Length}\;\mathrm{of}\;\mathrm{specimen}}$$

The precompression strain can be controlled by a fixed compressive load rather than by variations in the environment (such as material properties or temperature). It is maintained by the fixing device of a jig manufactured for research. The group of specimens having equal compressive stiffness and different specimen sizes is called Type 1 and that of specimens having different compressive stiffness is called Type 2. The fabricated specimens are shown in Figure 3, and the detailed design values and properties of all the specimens are listed in Table 1 and Table 2. In the case of the Type 1 specimen, the compressive stiffness ($k_{c}$) was marginally different from 5.52 kN/mm at the minimum to 5.96 kN/mm at the maximum. This is because, based on Equation (1), an approximate value was used to create a similar stiffness by adjusting the outer diameter and length of the specimen. In addition, Type 2 specimens with different compressive stiffnesses were manufactured to have a compressive stiffness increase rate of approximately 14.57%.

## 3. Experimental Set-Up

The cyclic loading test of polyurethane spring specimens was performed for three cycles using a universal testing machine (UTM). The displacement was measured using a linear variable displacement transducer (LVDT), and the compressive force was measured from a UTM load cell. The maximum strain was set in the range of 25–40%, and the loading speed ($v_{L}$) was set to 0.5 mm/s. The precompression strain was set to 7%, 14%, and 21% (see Table 3). In this study, the minimum precompression strain that can clearly verify the relationship between the precompression strain and recentering force through a sufficient number of preliminary experiments was calculated to be 7%.

Precompression can be generated by compressing a polyurethane spring specimen in advance with a precompression strain suitable for the experimental conditions and then, tightening it with a fixing nut to maintain the precompression state. As shown in Figure 4, the initial condition can be set by compressing the specimen to an extent equivalent to the strain corresponding to the precompression strain condition using the UTM and maintaining the corresponding compression displacement with a fixing nut. Cyclic loading tests were performed for the remaining displacements except for the precompression strain at the maximum strain. For example, in the case of a 100 L-92D-21P-30% specimen, a cyclic loading test was performed on 9% of the total length of the specimen (9 mm), except for the precompression strain of 21% (21 mm) at a maximum strain of 30% (30 mm), by imparting an initial precompression strain of 21%.

The loading condition was three cycles at a loading speed ($v_{L}$) of 0.5 mm/s through displacement control. The general loading protocol applied to the experiment is shown in Figure 5. A cycle is defined as the process of loading to the maximum strain and removing the load to attain the initial state. Accordingly, the time period (T) can be generalized to two times the loading speed and maximum displacement corresponding to the experimental conditions. The test time for a specimen is three times the total time period. However, the specimen under precompression behaves as much as the remaining strain except for the precompression strain at the maximum strain. The initial strain is assumed to be zero, and the maximum strain is calculated as the residual strain.

## 4. Experimental Results and Discussion

### 4.1. Behavior of Polyurethane Springs without Precompression

The force–displacement curve of the polyurethane spring under the conditions of five specimen sizes, a maximum strain of 30%, and precompression of 0% (representing the compression behavior of the specimen) is shown in Figure 6. The compression force (F) of the specimen shows significant strength degradation during the second loading after the first cycle. It can be observed that the unloading path shows almost similar behavior in all the cycles. In addition, it is verified that the second cycle after the first cycle was stabilized without significant stress reduction. Residual displacement occurs at the point where the unloading curve attains zero compressive force in each cycle and is restored to a certain level before the start of loading in the next cycle. This is a result of the fact that the delayed speed until the polyurethane spring is restored after compression is less than the loading speed of 0.5 mm/s. Figure 6a shows that the strength degradation after the first cycle is remarkably different from that of the remaining cases. Once the polyurethane spring is initially subjected to a certain level of external force, it is not restored fully, and internal variations occur. This implies that an identical result cannot be obtained.

The size of the Type 1 polyurethane spring specimen can be expressed based on the length of the specimen because it calculated the diameter with the same stiffness based on the length. That is, the larger the length of the specimen, the larger the size. Figure 7 summarizes the trends of maximum stress corresponding to maximum strain values of 25%, 30%, 35%, and 40%. The maximum stress tended to be inversely proportional to the length of the specimen. Linear regression analysis revealed a linear relationship with $R^{2}$ of 0.9179, 0.9103, 0.9148, and 0.8898, respectively. This implies that the variation in the cross-sectional area of the polyurethane spring was adjusted to maintain the compressive stiffness. It increases by an amount larger than the increase in the force corresponding to the maximum strain. Therefore, a polyurethane spring with identical compressive stiffness can resist larger stress as its size decreases. While designing a damper or seismic isolator with a polyurethane spring, it is necessary to adjust the compressive stiffness and specimen size to satisfy the design conditions. This aspect should be considered while designing a damper or seismic isolator with a polyurethane spring, and the compressive stiffness and specimen size suitable for the design conditions must be attained.

The compressive stiffness of the polyurethane spring is inversely proportional to the length of the specimen. For Type 2, the compressive stiffness was adjusted by fixing the outer diameter to 100 mm and then, decreasing the length of the specimen from 100 mm to 60 mm in increments of 10 mm. Therefore, the variation in the length of the Type 2 specimen can be represented by the variation in the compressive stiffness. Figure 8 shows the trend of the maximum stress corresponding to maximum strain values of 25%, 30%, 35%, and 40%. Under all the experimental conditions, the maximum stress tends to be proportional to the compressive stiffness. The linear regression analysis verified the existence of a linear relationship with $R^{2}$ values of 0.9233, 0.9224, 0.9127, and 0.8949, respectively. This implies that the relationship between the compressive stiffness of the polyurethane spring and the maximum stress can be linearly idealized. It can also be observed that the slope of the regression equations of compressive stiffness and maximum stress increases gradually as the maximum strain increases. Therefore, when a larger deformation occurs, the higher the compressive stiffness, the larger the increase in the stress required for deformation.

### 4.2. Behavior of Precompressed Polyurethane Springs

Polyurethane springs show a flag-shaped hysteresis behavior similar to shape memory alloys with additional restoring force under precompression conditions. Figure 9 schematically explains the precompression behavior of a polyurethane spring. In this study, the initial restoring force generated by precompression is defined as the recentering force ($F_{r}$). It refers to the internal force remaining in the specimen when the displacement in the unloading path becomes zero.

Figure 10 shows the case of the 100 L-92D-30% specimen representing the cyclic loading test results of polyurethane springs according to the precompression strain. Each specimen was pre-applied with a compressive force corresponding to 7%, 14%, and 21% of the precompression strain. The results under an identical maximum strain condition are shown. As can be observed in Figure 10b–d, the polyurethane spring under precompression conditions shows the recentering force. This force tends to be proportional to the precompression strain.

Similar to the polyurethane spring without precompression, the precompressed polyurethane spring also shows a considerable decrease in strength after the first cycle (see Figure 11). In addition, the unloading curve shows a similar path regardless of each cycle. It can be observed that the loading curve stabilizes with a significant decrease from the second loading. The area of the force–displacement curve of the precompressed polyurethane spring shows a behavior similar to that of the specimen without precompression. After the first cycle, the area of the closed curve shows a significant decrease in the second cycle. It is verified that the areas of the second and third cycles do not differ significantly and are stable. The recentering force decreases marginally as the loading cycle is repeated. The fact that the recentering force is the internal force generated by the initial precompression, implies that the restoring force can be maintained without being reduced substantially by using the external force.

The results of the regression analysis of the recentering force of the polyurethane spring according to the variation in precompression strain are summarized in Figure 12 and Table 4. Figure 12 shows the results of the regression analysis of the Type 1 specimen. Table 3 shows the results of the regression analysis of Type 2. This is because the recentering force shows similar results regardless of the variation in the compressive stiffness and size of the specimen. The relationship between the precompression strain and recentering force tends to increase proportionally in the range of 7% to 21%. From these results, it is feasible to determine the precompression strain at which the recentering force becomes zero by using the linear regression equation of the precompression strain and the recentering force. This value of precompression strain corresponds to the x-intercept of the regression equation. It represents the minimum precompression strain that does not cause residual displacement of the polyurethane spring. Therefore, the restoring force required when the polyurethane spring is applied in the system as a compression member can be calculated by estimating the precompression strain that can remove the residual displacement.

Figure 13 shows the force–displacement graph from 0% to 21% of the precompression strain of the Type 2 specimen with a maximum strain of 40% in the graph. It can be observed that the recentering force corresponding to each precompression strain is located on the unloading path of the polyurethane spring without precompression. This tendency indicates that the recentering force of the polyurethane spring under precompression conditions can be predicted relatively straightforwardly based on the compression behavior of the polyurethane spring without precompression. Thus, the recentering force can be estimated based on the compression behavior of a general polyurethane spring and utilized in the design of the restoring force of the system.

It can be observed that similar to the tendency of the precompression behavior of the polyurethane spring, the recentering force generated by precompression is distributed over the unloading path of the general compression behavior without precompression conditions. Therefore, a predictive model was established based on this tendency. It can predict the recentering force corresponding to any precompression strain (see Figure 14).

The predictive model was verified by comparing the recentering force measured through the experiment and the recentering force corresponding to the following values of the strain of the polyurethane spring without precompression conditions: 7%, 14%, and 21%. As shown in Figure 15 and Table 5, the linear regression analysis of the measured and predicted recentering forces verifies that the measured value is smaller than the predicted value. There is a significant difference between the predicted and measured values at the precompression strain of 7%. The measured value of the recentering force at the precompression strain of 14% is smaller than the predicted value by 24% on an average. Furthermore, the result is 11% lower on average at the precompression strain of 21%. Thus, the recentering force obtained using the predictive model reviewed in this study is significantly different from the actual measured recentering force. However, additional research is required on the reduction rate that can supplement this through the tendency of the recentering force to be distributed near the unloading curve and the tendency of the measured value to be generally lower. The reduction rate thus obtained can be applied to estimate the recentering force of the polyurethane spring as an accurate predictive model.

## 5. Conclusions

In this study, cyclic loading tests were performed to understand the behavioral characteristics of polyurethane springs. These exhibited behaviors were similar to those of shape memory alloys under precompression conditions and displayed recentering characteristics. A total of 160 polyurethane spring specimens were designed to evaluate the effect of the design variables on the compression behavior of the polyurethane springs. In addition, a predictive model based on the tendency of the recentering force was established and evaluated. The following are the main conclusions:

The polyurethane spring showed a significant decrease in strength after the first of the cyclic loads and the tendency to behave stably from the second cycle was verified. The maximum compressive force of the polyurethane spring was decreased by cycle repetition, although the difference was highly marginal. For polyurethane springs of different sizes with identical compressive stiffness, the maximum stress tended to decrease as the size of the specimen increased. In addition, it was verified that the slope of the linear regression equation increased gradually as the maximum strain increased. It is observed that when a deformation of at least 25% of the total is likely, the higher the compressive stiffness, the higher the efficiency of resistance against the external force.

It was verified that the recentering force caused by the precompression of the polyurethane spring had a linear relationship with the precompression strain. Therefore, the recentering force can be calculated by estimating the precompression strain that can minimize the residual displacement. The distribution trend of the recentering force indicates the predictability of the recentering force based on the compression behavior of general polyurethane springs. However, the measured value tends to be less than the predicted value. Therefore, additional research is required to reduce the error.

## Figures and Tables

**Figure 1 materials-15-03514-f001:**
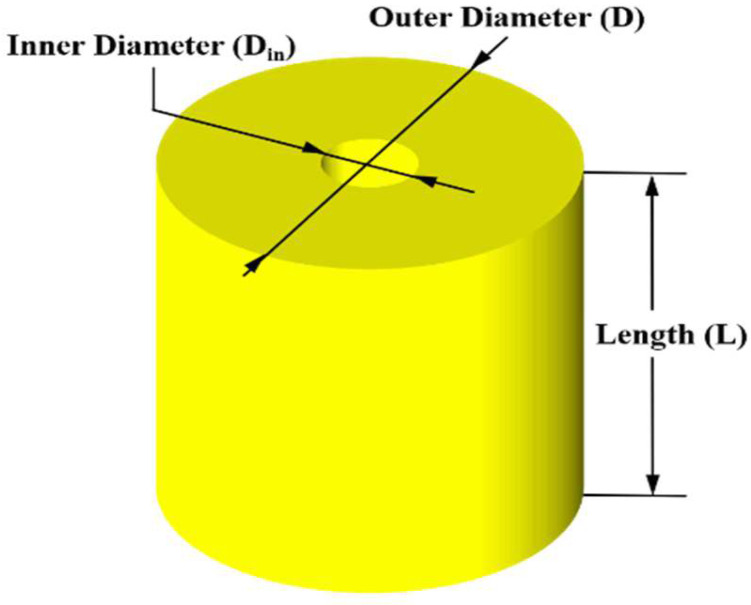
Design of Polyurethane springs.

**Figure 2 materials-15-03514-f002:**
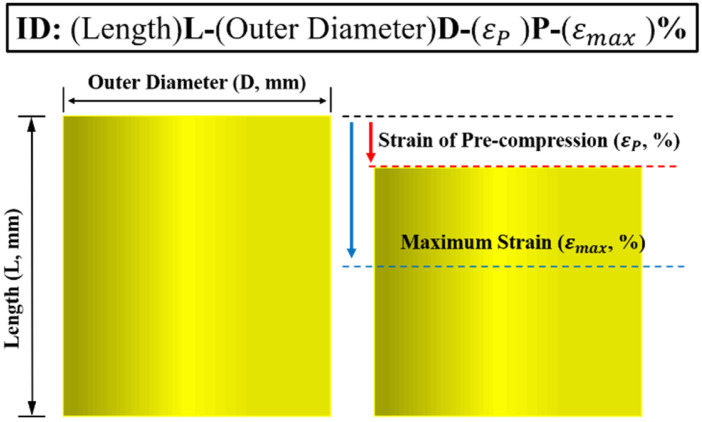
Explanation of specimen identification.

**Figure 3 materials-15-03514-f003:**
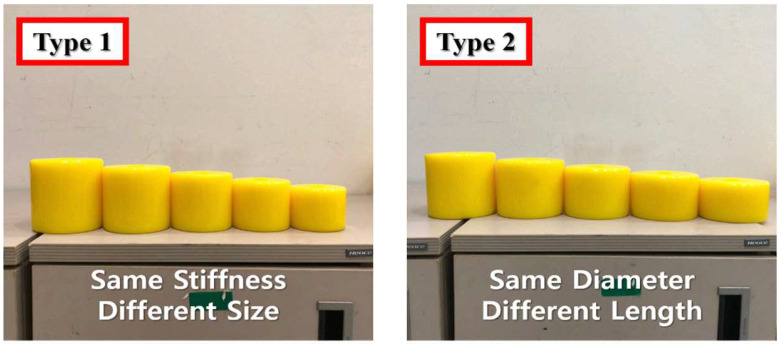
Manufactured specimens of each type.

**Figure 4 materials-15-03514-f004:**
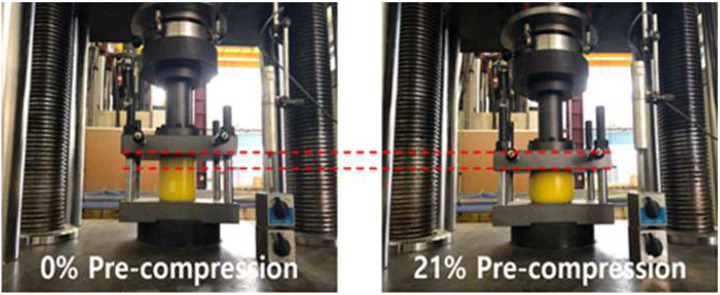
Method of setting initial conditions of precompressed polyurethane springs.

**Figure 5 materials-15-03514-f005:**
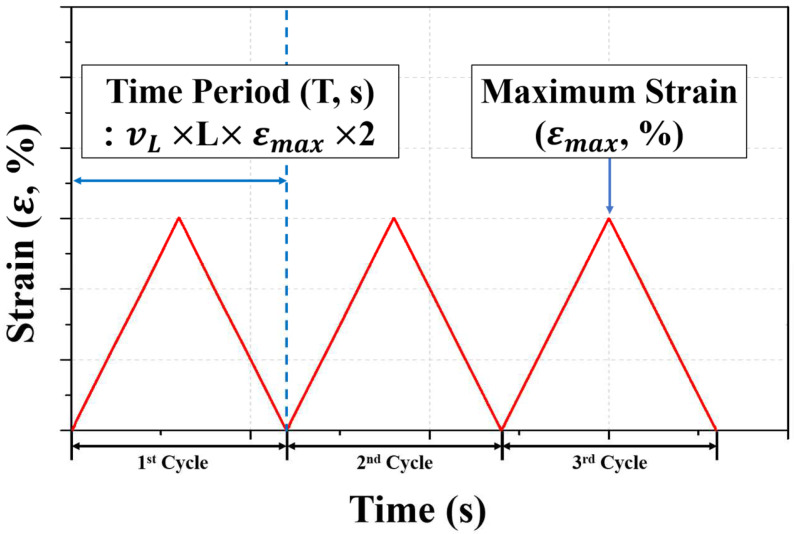
Loading protocol of cyclic loading test.

**Figure 6 materials-15-03514-f006:**
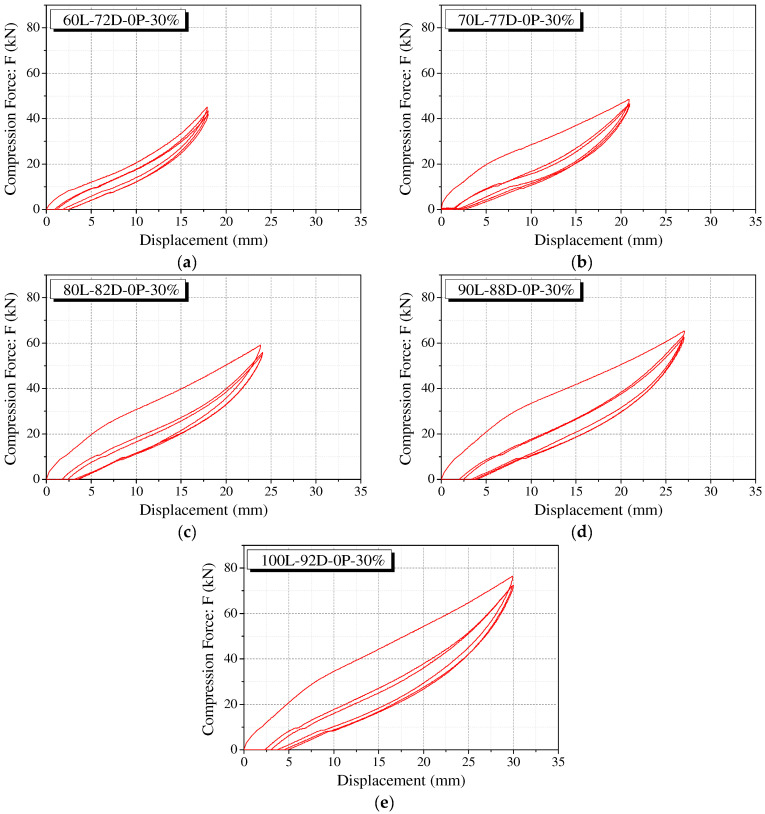
Compression behavior of polyurethane springs without precompression. (**a**) 60L-72D-0P-30%; (**b**) 70L-77D-0P-30%; (**c**) 80L-82D-0P-30%; (**d**) 90L-88D-0P-30%; (**e**) 100L-92D-0P-30%.

**Figure 7 materials-15-03514-f007:**
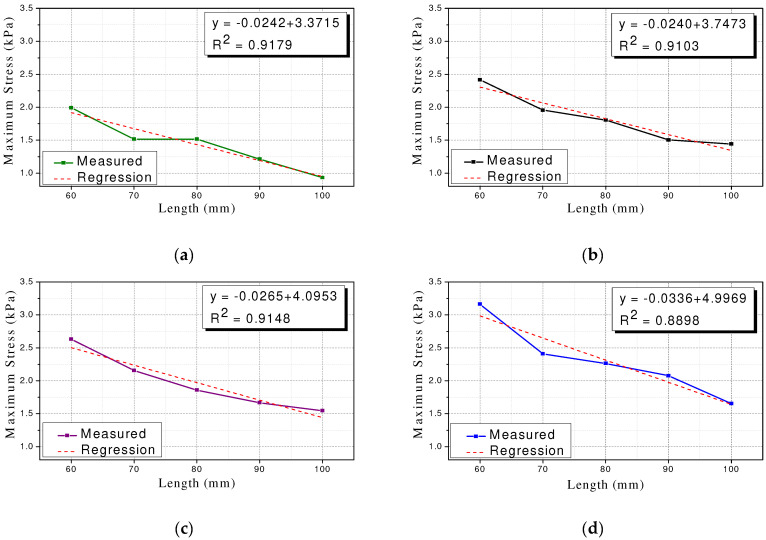
Relationship between specimen size and maximum stress (Type 1). (**a**) Type 1–25%; (**b**) Type 1–30%; (**c**) Type 1–35%; (**d**) Type 1–40%.

**Figure 8 materials-15-03514-f008:**
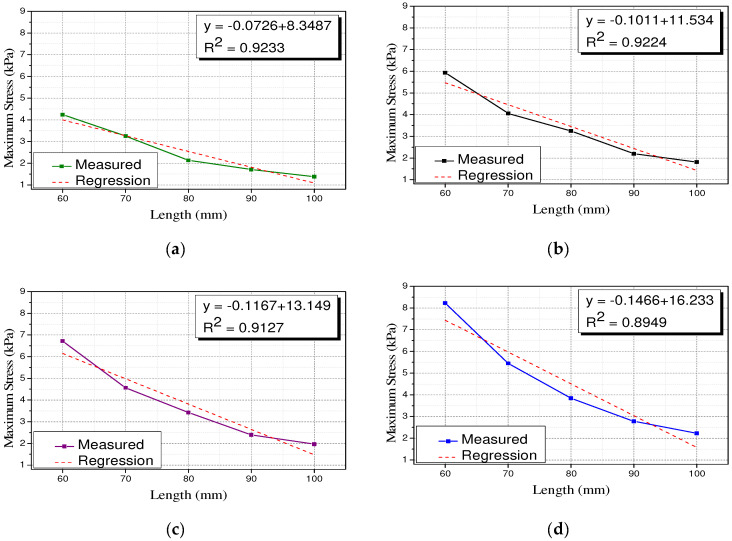
Relationship between compressive stiffness and maximum stress (Type 2). (**a**) Type 2–25%; (**b**) Type 2–30%; (**c**) Type 2–35%; (**d**) Type 2–40%.

**Figure 9 materials-15-03514-f009:**
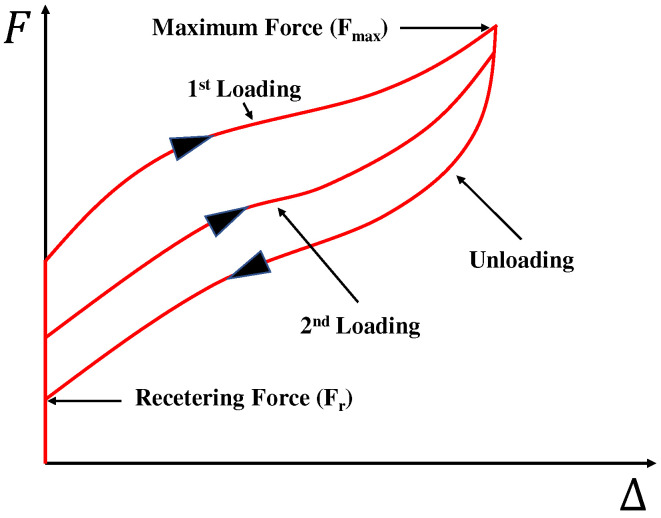
Behavior of precompressed polyurethane spring.

**Figure 10 materials-15-03514-f010:**
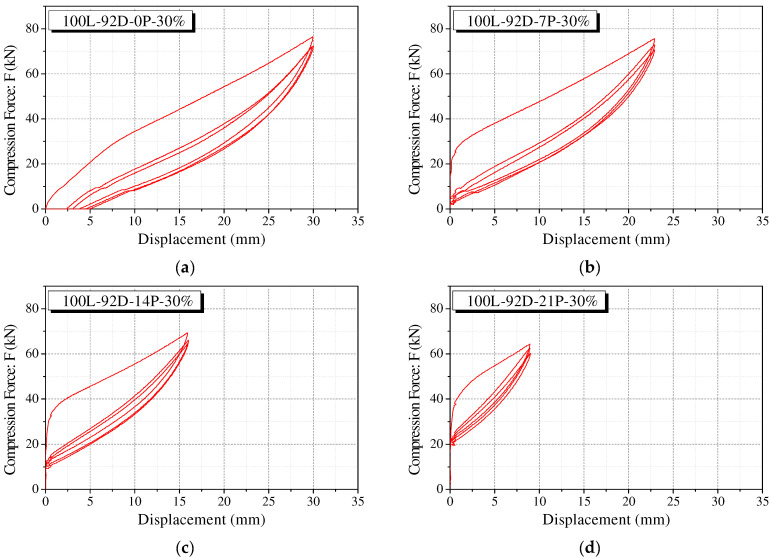
Cyclic behavior of precompressed polyurethane spring. (**a**) 100L-92D-0P-30%; (**b**) 100L-92D-7P-30%; (**c**) 100L-92D-14P-30%; (**d**) 100L-92D-21P-30%.

**Figure 11 materials-15-03514-f011:**
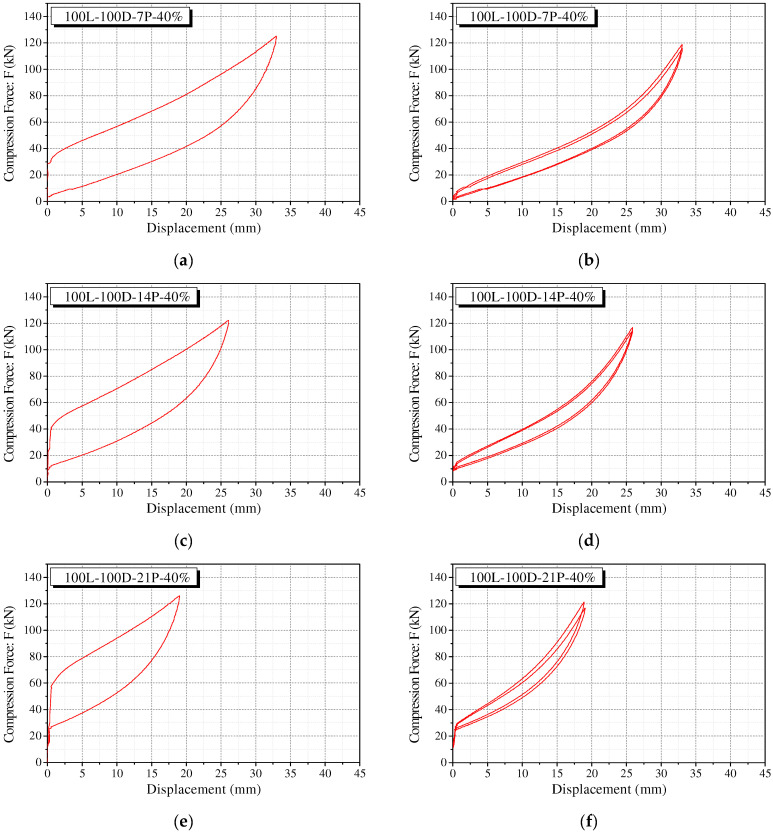
Behavior of precompressed polyurethane springs in separated cycle. (**a**) 100L-100D-7P-40% at 1st cycle; (**b**) 100L-100D-7P-40% after 1st cycle; (**c**) 100L-100D-14P-40% at 1st cycle; (**d**) 100L-100D-14P-40% after 1st cycle; (**e**) 100L-100D-21P-40% at 1st cycle; (**f**) 100L-100D-21P-40% after 1st cycle.

**Figure 12 materials-15-03514-f012:**
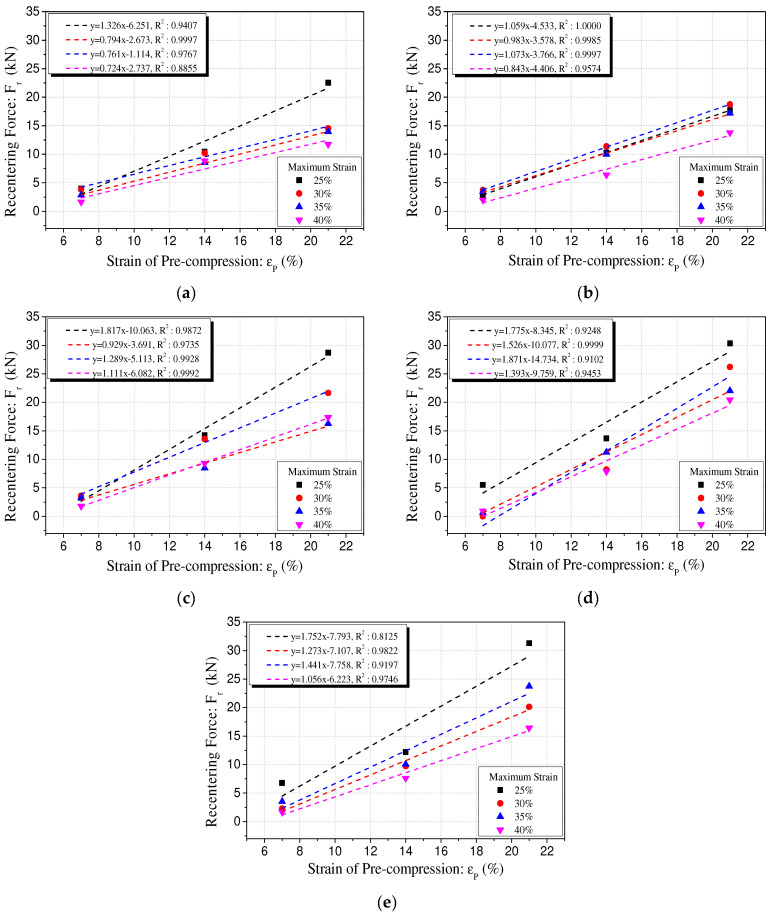
Regression analysis of recentering force with precompression strain [Type 1]. (**a**) 60L-72D; (**b**) 70L-77D; (**c**) 80L-82D; (**d**) 90L-88D; (**e**) 100L-92D.

**Figure 13 materials-15-03514-f013:**
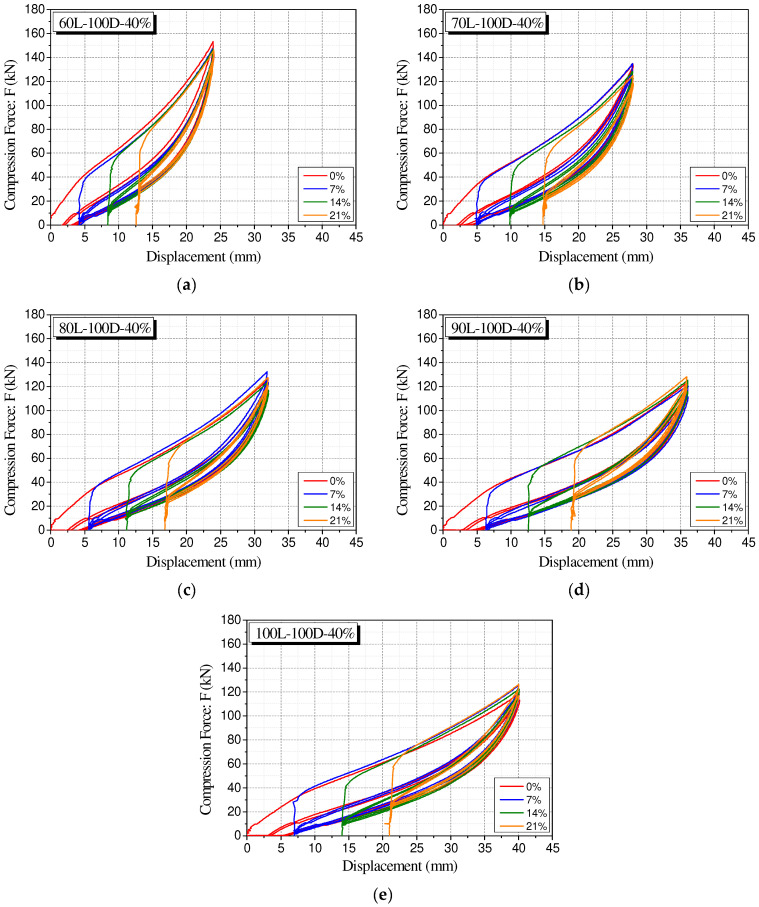
Integrated force-displacement curves of all experimental conditions. (**a**) 60L-100D-40%; (**b**) 70L-100D-40%; (**c**) 80L-100D-40%; (**d**) 90L-100D-40%; (**e**) 100L-100D-40%.

**Figure 14 materials-15-03514-f014:**
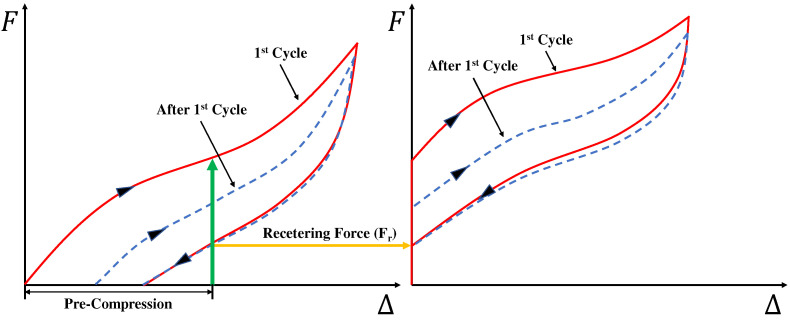
Descriptions of recentering force prediction model.

**Figure 15 materials-15-03514-f015:**
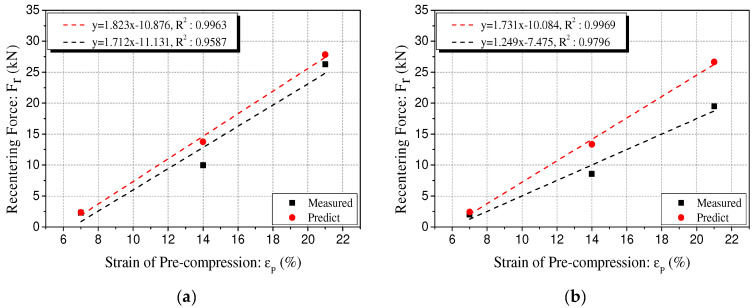
Comparison of measured recentering force and predicted recentering force. (**a**) 60L-100D-40%; (**b**) 70L-100D-40%; (**c**) 80L-100D-40%; (**d**) 90L-100D-40%; (**e**) 100L-100D-40%.

**Table 1 materials-15-03514-t001:** Properties of both the types of polyurethane springs.

Type	D (mm)	D_in_ (mm)	L (mm)	S	E_a_ (MPa)	k_c_ (kN/mm)
1	92	20	100	0.18	87.21	5.52
	88	20	90	0.20	88.03	5.85
	82	20	80	0.20	88.37	5.72
	77	20	70	0.21	89.03	5.80
	72	20	60	0.23	89.97	5.96
2	100	20	100	0.21	88.57	6.87
	100	20	90	0.23	89.94	7.75
	100	20	80	0.26	91.86	8.90
	100	20	70	0.29	94.65	10.48
	100	20	60	0.34	98.95	12.79

**Table 2 materials-15-03514-t002:** Material properties of polyurethane springs (TDI/PTMG based prepolymer).

Typical Prepolymer Properties	
% NCO	6.1–6.5
Brookfield Viscosity @ 100 °C	1.5–4.5
Specific Gravity	
@ 25 °C	1.07
@ 100 °C	1.01
Typical Physical Properties	
Shore Hardness	95A
100% Modulus, psi (MPa)	1900 (13.0)
300% Modulus, psi (MPa)	4000 (27.5)
Tensile, psi (MPa)	5500 (37.9)
Elongation, %	400
Tear Strength, Die C, pli (kN/m)	550 (96.0)
Tear Strength, (D470), pli (kN/m)	140 (24.5)
Bashore Rebound, %	40
Compression Set, Method B, 22 h @ 70 °C, %	38
Specific Gravity	1.13

**Table 3 materials-15-03514-t003:** Experimental conditions.

Condition				
Precompression Strain (ε*_P_*)	0%	7%	14%	21%
Maximum Strain (ε*_max_*)	25%	30%	35%	40%
Loading Speed (v_L_)	0.5 mm/s			
Loading Cycle	3 cycle			

**Table 4 materials-15-03514-t004:** Regression results between recentering force and precompression strain.

Type	Specimen	Slop	Y-Intercept	R^2^	X-Intercept
1	100L-92D-25%	1.752	−7.793	0.8125	4.45
	90L-88D-25%	1.775	−8.345	0.9248	4.70
	80L-82D-25%	1.817	−10.063	0.9872	5.54
	70L-77D-25%	1.059	−4.533	1	4.28
	60L-72D-25%	1.326	−6.251	0.9407	4.71
	100L-92D-30%	1.273	−7.107	0.9822	5.58
	90L-88D-30%	1.526	−10.077	0.9999	6.60
	80L-82D-30%	0.929	−3.691	0.9735	3.97
	70L-77D-30%	0.983	−3.578	0.9985	3.64
	60L-72D-30%	0.794	−2.673	0.9997	3.37
	100L-92D-35%	1.441	−7.758	0.9197	5.38
	90L-88D-35%	1.871	−14.734	0.9102	7.87
	80L-82D-35%	1.289	−5.113	0.9928	3.97
	70L-77D-35%	1.073	−3.766	0.9997	3.51
	60L-72D-35%	0.761	−1.114	0.9767	1.46
	100L-92D-40%	1.056	−6.223	0.9746	5.89
	90L-88D-40%	1.393	−9.759	0.9453	7.01
	80L-82D-40%	1.111	−6.082	0.9992	5.47
	70L-77D-40%	0.843	−4.406	0.9574	5.23
	60L-72D-40%	0.724	−2.737	0.8855	3.78
2	100L-100D-25%	2.6504	−12.616	1	4.76
	90L-100D-25%	2.5716	−13.26	0.91478	5.16
	80L-100D-25%	2.5716	−14.978	0.7571	5.82
	70L-100D-25%	3.1399	−23.634	0.8276	7.53
	60L-100D-25%	3.4823	−21.159	0.9833	6.08
	100L-100D-30%	1.6934	−7.6481	0.9865	4.52
	90L-100D-30%	2.3231	−16.463	0.9995	7.09
	80L-100D-30%	2.3163	−12.485	0.9721	5.39
	70L-100D-30%	1.8094	−7.5633	0.9948	4.18
	60L-100D-30%	2.189	−9.3453	0.9957	4.27
	100L-100D-35%	1.8245	−9.4762	0.9965	5.19
	90L-100D-35%	1.9882	−13.026	0.9705	6.55
	80L-100D-35%	1.7806	−8.4791	0.9954	4.76
	70L-100D-35%	1.9003	10.197	0.9504	-5.37
	60L-100D-35%	2.0473	11.598	0.9868	-5.67
	100L-100D-40%	1.5851	−10.304	0.9438	6.50
	90L-100D-40%	1.6669	−10.388	0.9966	6.23
	80L-100D-40%	1.4669	−8.9317	0.9789	6.09
	70L-100D-40%	1.2487	−7.4749	0.9796	5.99
	60L-100D-40%	1.7124	−11.131	0.9587	6.50

**Table 5 materials-15-03514-t005:** Regression relationship between predicted recentering force and precompression strain.

Type	Specimen	Slop	Y-Intercept	R^2^	X-Intercept
1	100L-92D-25%	1.843	−6.6404	0.9927	3.60
	90L-88D-25%	2.01	−9.4479	0.989	4.70
	80L-82D-25%	1.7624	−5.7635	0.9897	3.27
	70L-77D-25%	1.3851	−5.7989	0.987	4.19
	60L-72D-25%	1.332	−5.1695	0.9861	3.88
	100L-92D-30%	1.858	−9.1579	0.9916	4.93
	90L-88D-30%	1.622	−6.9445	0.9969	4.28
	80L-82D-30%	1.4911	−6.8808	0.9975	4.61
	70L-77D-30%	1.1229	−3.9531	0.9952	3.52
	60L-72D-30%	1.088	−4.5613	0.9933	4.19
	100L-92D-35%	1.605	8.3588	0.9967	−5.21
	90L-88D-35%	1.379	−6.3929	0.9985	4.64
	80L-82D-35%	1.129	−5.6716	0.9993	5.02
	70L-77D-35%	1.0047	−4.9644	0.9994	4.94
	60L-72D-35%	0.9517	−4.4906	0.999	4.72
	100L-92D-40%	1.4101	−8.9104	0.9962	6.32
	90L-88D-40%	1.2472	−6.4777	0.9999	5.19
	80L-82D-40%	1.0941	−5.6928	0.9999	5.20
	70L-77D-40%	0.8759	−3.9461	0.9987	4.51
	60L-72D-40%	0.835	−3.1894	0.9993	3.82
2	100L-100D-25%	1.8427	−6.6121	0.9934	3.59
	90L-100D-25%	2.7337	−11.654	0.9904	4.26
	80L-100D-25%	2.5519	−9.7873	0.9898	3.84
	70L-100D-25%	2.9217	−11.279	0.9911	3.86
	60L-100D-25%	2.8141	−12.22	0.9882	4.34
	100L-100D-30%	2.3003	−11.958	0.9957	5.20
	90L-100D-30%	2.2822	−11.463	0.9955	5.02
	80L-100D-30%	2.4034	−9.2994	0.9923	3.87
	70L-100D-30%	2.2867	−11.527	0.9904	5.04
	60L-100D-30%	2.4958	−12.298	0.9903	4.93
	100L-100D-35%	1.8791	−9.7873	0.9965	5.21
	90L-100D-35%	1.9185	−10.431	0.9965	5.44
	80L-100D-35%	1.9912	−9.9005	0.9973	4.97
	70L-100D-35%	1.8336	−7.885	0.9958	4.30
	60L-100D-35%	2.0442	−8.8044	0.9954	4.31
	100L-100D-40%	1.623	−9.1792	0.9989	5.66
	90L-100D-40%	1.6654	−9.6318	0.9985	5.78
	80L-100D-40%	1.6563	−9.7237	0.9986	5.87
	70L-100D-40%	1.7306	−10.084	0.9969	5.83
	60L-100D-40%	1.823	−10.876	0.9963	5.97

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
