# Peer review of "Experimental Study on Recentering Behavior of Precompressed Polyurethane Springs"

_materials, 2022, doi:10.3390/ma15103514_

Round 1

Reviewer 1 Report

The manuscript is devoted to the experimental investigation of the recentering behaviour of precompressed PU springs adopted in seismic isolators and dampers. A simple model was proposed to relate the recentering force with the precompression strain.

I find the paper really interesting, but few observations are necessary:

  1. In the introduction the authors say that the novelty of the work lies in the proposition of a prediction model for the recentering force of PU spring. Please provide an overview of other possible model for force recentering and highlight the differences with the one proposed by the authors. Enhance the originality and contribution of your work respect the current literature because it is not very clear.
  2. Section 2, equation 2-3, add the reference within the text for these equations. Explain better the concept of the Shape factor S.
  3. Page 4: “…because the outer diameter was adjusted considering the convenience of specimen production”. Please, explain better the why the outer diameter was adjusted. For the same reason, better explain at page 7 “The size of the Type 1 polyurethane spring specimen can be expressed based on the length of the specimen because it was manufactured by adjusting the diameter based on the length for convenient production”.
  4. Figure 13: express the meaning of the legend.
  5. Conclusion: the authors state that “Hence, it can resist a constant maximum compressive force even under cyclic loads”. This reviewer think that 4 cycles test are too low to say that. Longer tests are required to verify the durability of the PU spring vs. the cyclic load.

For all the previous reasons, the reviewer recommends Minor amendments of paper for publication in Materials.

Author Response

Dear Reviewer #1

Please check the attached file. Thank you.

Jong Wan Hu

Reviewer 2 Report

Dear Editor and Authors,

I read carefully the paper and I can conclude that is an interesting and applicative one. Before publication in the journal some issues must be clarified:

  1. Please add in the material section the type of polyurethane-elastomer used.
  2. Please add the errors bar in Figures 7 and 8, for the maximum stress measured.
  3. Check the English.

Author Response

Dear Reviewer #2

Please check the attached file. Thank you.

Jong Wan Hu

Reviewer 3 Report

The paper "Experimental Study on Recentering Behavior of
Precompressed Polyurethane Springs" presents a topic of interest to potential readers of Materials, however there is a lack of innovation aspects that could effectively justify the publication of this paper. The authors should highlight this more clearly to readers in several parts of the text, in addition we have:

(1) The abstract is incomplete, lacks clear information on results and highlights the author's main findings.
(2) The theoretical reference is very old, note that only in the introduction there are 24 references, and in the rest of the text only two more, this is not admitted;
(3) The level of discussion is superficial, few relevant and current references are used, the discussions need to be clearly deepened by the authors, comparing their results with other studies in the literature;
(4) The conclusion needs reformulation, it is long, I suggest something around 3 paragraphs maximum.

The points highlighted above are very relevant and important, which prevent me at this moment from accepting the paper, so I am for its rejection, encouraging the authors to resubmit their paper after the corrections.

Author Response

Dear Reviewer # 3

Please check the attached file. Thank you.

Jong Wan Hu

Reviewer 4 Report

  1. Spelling mistake in the title of the article: Poliurethane à           Page 1
  2. Fabrication process of polyurethane springs is missing.
  3. Redefine pre-compression strain (ε?) according to the formula. i.e. is the ratio of the linear compression displacement to the total specimen length not vice-versa Eq 4, Page 4
  4. Repetitive sentence for Fig. 12 Table 3.         page 11
  5. What about the reliability (Useful life) of polyurethane springs under seismic conditions?

Author Response

Dear Reviewer # 4

Please check the attached file. Thank you.

Jong Wan Hu

Round 2

Reviewer 2 Report

Accept

Reviewer 3 Report

The authors made all the suggested corrections.